# Functional Janus structured liquids and aerogels

**Ahmadreza Ghaffarkhah[1,2,10], Seyyed Alireza Hashemi[1,10], Farhad Ahmadijokani[1,2], Milad Goodarzi[1], Hossein Riazi[3], Sameer E. Mhatre[2], Orysia Zaremba[4], Orlando J. Rojas [2], Masoud Soroush [3], Thomas P. Russell [5,6,7] ✉, Stefan Wuttke [4,8] ✉, Milad Kamkar [9] ✉ & Mohammad Arjmand [1] ✉**

Janus structures have unique properties due to their distinct functionalities on opposing faces, but have yet to be realized with flowing liquids. We demonstrate such Janus liquids with a customizable distribution of nanoparticles (NPs) throughout their structures by joining two aqueous streams of NP dispersions in an apolar liquid. Using this anisotropic integration platform, different magnetic, conductive, or non-responsive NPs can be spatially confined to opposite sides of the original interface using magnetic graphene oxide (mGO)/GO, $Ti_3C_2T_x$/GO, or GO suspensions. The resultant Janus liquids can be used as templates for versatile, responsive, and mechanically robust aerogels suitable for piezoresistive sensing, human motion monitoring, and electromagnetic interference (EMI) shielding with a tuned absorption mechanism. The EMI shields outperform their current counterparts in terms of wave absorption, i.e., $SE_T \approx 51$ dB, $SE_R \approx 0.4$ dB, and A = 0.91, due to their high porosity ranging from micro- to macro-scales along with non-interfering magnetic and conductive networks imparted by the Janus architecture.

The self-assembly and jamming of nanoparticles (NPs) at the liquid–liquid interface is an emerging platform for the fabrication of soft functional materials, where the inherent properties of NPs can be integrated into the final structures[1–3]. In practice, interfacial jamming can be realized when the functionalized NPs dispersed in one liquid interact with polymeric/oligomeric ligands with complementary functionality dissolved in a second immiscible liquid, forming what has been termed NP surfactants (NPSs)[1,4–6]. This process arrests the reduction in the interfacial area, stabilizing highly non-equilibrium shapes of liquids.

Due to the nature of jamming, external stimuli can be used to control the collective properties via manipulating the binding of the NPSs to the interface, imparting responsiveness to the final constructs[7,8]. Reconfigurable on-demand, interfacial jamming allows the fabrication of novel all-liquid systems, e.g., reconfigurable discrete droplets[9,10], molded liquid objects[11], and bicontinuous interfacially jammed emulsion gels (bijels)[12], with unique characteristics and applications.

Liquid streaming is a new process where a jetted stream of one liquid in an external immiscible liquid can be stabilized by interfacial

[1]Nanomaterials and Polymer Nanocomposites Laboratory, School of Engineering, University of British Columbia, Kelowna, BC V1V 1V7, Canada. [2]Bioproducts Institute, Department of Chemical & Biological Engineering, Department of Chemistry and Department of Wood Science, The University of British Columbia, 2360 East Mall, Vancouver, BC V6T 1Z3, Canada. [3]Department of Chemical and Biological Engineering, Drexel University, Philadelphia, PA 19104, USA. [4]Basque Center for Materials, Applications and Nanostructures (BCMaterials), Bld. Martina Casiano, 3rd Floor UPV/EHU Science Park Barrio Sarriena s/n, 48940 Leioa, Spain. [5]Polymer Science and Engineering Department, University of Massachusetts Amherst, 120 Governors Drive, Amherst, MA 01003, USA. [6]Materials Sciences Division, Lawrence Berkeley National Laboratory, 1 Cyclotron Road, Berkeley, CA 94720, USA. [7]Advanced Institute for Materials Research (WPI-AIMR), Tohoku University, 2-1-1 Katahira, Aoba, Sendai 980-8577, Japan. [8]IKERBASQUE, Basque Foundation for Science, 48013 Bilbao, Spain. [9]Department of Chemical Engineering, Waterloo Institute for Nanotechnology, University of Waterloo, 200 University Avenue West, Waterloo, ON N2L 3G1, Canada. [10]These authors contributed equally: Ahmadreza Ghaffarkhah, Seyyed Alireza Hashemi. ✉e-mail: russell@mail.pse.umass.edu; stefan.wuttke@bcmaterials.net; milad.kamkar@uwaterloo.ca; mohammad.arjmand@ubc.ca

jamming, forming tubular liquid threads[1,5,13]. For the efficient generation of these tubular all-liquid structures, the rate of NPSs formation, assembly, and jamming must be faster than the relaxation of fluids, while their binding energy must be high enough to impart mechanical stability[14–16]. This process generates a robust skin around the tubular jetted liquid threads that maintains the integrity of this class of structured liquids[1,5,15]. Taking advantage of liquid jetting, several new concepts of structured liquids, e.g., all-liquid printing[15–17] and 3D water in oil tubular emulsions, with tunable morphology and domain size[5], were put forward. But, so far, little advantage has been taken from the combination of functional NPs and liquid streaming to generate constructs with a specific function in soft robotics[18], microfluidics[15,19,20], or sensing[17,21].

Here, we report a one-step fabrication of Janus structured liquids with an anisotropic distribution of NPs. Our approach relies on stabilizing and then merging two aqueous streams of NPs in a nonpolar liquid containing complementary functionalized ligands. The resulting liquid threads consist of two distinct sections, each composed of specific groups of NPs. This unique duality inherent in Janus liquids affords an opportunity to assign distinct functionalities to opposing sides of the structures and fine-tune them independently. As a proof of concept and to highlight the future potential of these unique constructs, magnetic, conductive, or non-responsive NPs were sequestered in opposite sides of these constructs using magnetic graphene oxide (mGO)/GO, $Ti_3C_2T_x$/GO, or GO suspensions, allowing the formation of multi-responsive soft materials that can be heterogeneously tailored from the micro- to macro-scale.

Such precise control over the functionality of Janus constructs becomes even more pronounced when these structured liquids are used as templates, enabling the development of customized compositions and arrangements for task-oriented aerogels. The potential applications of Janus aerogels are vast, especially when a deliberate connection is established between the functionality of the Janus building blocks and the ultimate application. For instance, the development of Janus aerogels with non-interfering magnetic/conductive opposing sections represents a breakthrough in electromagnetic interference (EMI) shielding. By integrating alternating magnetic/conductive domains within the aerogel structure, Janus constructs surpass existing shielding materials, offering superior wave absorption properties. This addresses a long-standing challenge in EMI shielding and positions Janus aerogels as highly promising candidates for next-generation shielding materials. This integration platform is also used to make functional aerogels flexible, where functionality, e.g., electrical conductivity, is customized through one side of the structures, and mechanical flexibility is mainly derived from the opposite side. This adjustable strategy opens up numerous opportunities in pressure sensing and human motion monitoring.

## Results and discussion
### Shaping GO non-responsive liquid threads

The fabrication of Janus liquids requires: (1) jetting separate liquid threads, (2) bringing the liquid threads together during jetting, (3) and merging the threads. Three different 2D NPs, including graphene oxide (GO), GO decorated with $Fe_3O_4$ or magnetic GO (mGO), and highly conductive $Ti_3C_2T_x$, were prepared, dispersed in water, and jetted (see Figs. S1–5 and the corresponding discussion in Supplementary Information regarding the synthesis and characterization of the 2D NPs).

We initially produced non-responsive liquid threads by streaming 5–10 mg/ml aqueous GO suspensions into hexane containing 1 mg/ml PSS-[3-(2-aminoethyl) amino]propyl-heptaisobutyl substituted POSS (POSS-NH$_2$) (Fig. 1a, S6a, b, and Video S1). Here, the rapid co-assembly of GO and POSS at the interface forms an interfacial skin that wraps the jetted liquid thread[5,16]. This interfacial skin prevents the jet from breaking up into droplets, suppressing Plateau–Rayleigh instabilities[1,4]. The interfacial layer imparts mechanical stability to these highly non-equilibrium-shaped aqueous threads, allowing them to remain intact even after being agitated or aged for months (Fig. S6b and Video S2).

The formation of the non-responsive liquid threads is based on the dual hydrophobic/hydrophilic nature of GO, i.e., the hydrophobic basal plane and hydrophilic functionality such as hydroxyl, carboxylic, and epoxy groups, which promotes the very rapid migration and assembly of these NPs at the oil/water interface[5,13,22]. When GO is in the aqueous phase, both sides of the hydrophobic basal planes are in contact with water; however, upon migration to and assembly at the interface, not only are the bare water oil contacts removed by the

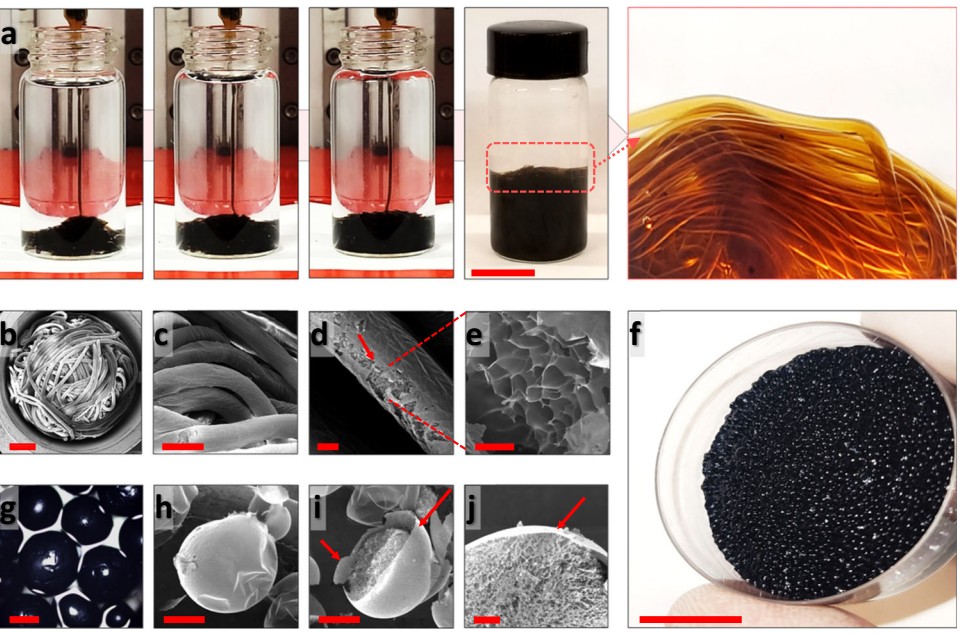

**Fig. 1 | Shaping GO non-responsive liquid threads and $Ti_3C_2T_x$ beads. a** Digital images demonstrating tubule formation upon streaming GO aqueous suspension (10 mg/ml) into hexane-POSS. **b–e** FESEM images of worm-like aerogels of GO. **f** Digital and **g** confocal images of $Ti_3C_2T_x$ beads. **h–j** FESEM images of $Ti_3C_2T_x$ aerogel beads. The red arrows in (**d**) and (**i, j**) highlight the laminated skin that covers the bulk of aerogels. Scale bars in (**a, f**), (**b**), (**c**), (**d, g**), (**h, i**), and (**e–j**) correspond to 2 cm, 2000, 1000, 500, 400, and 100 μm, respectively.

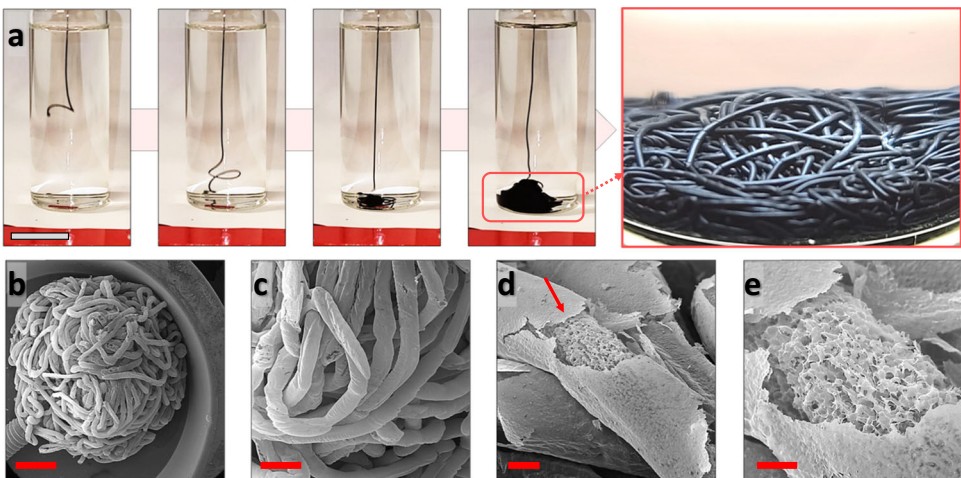

**Fig. 2 | Shaping Ti₃C₂Tₓ/GO worm-like constructs. a** Digital images demonstrating tubule formation upon streaming 10 mg/ml Ti₃C₂Tₓ/GO aqueous containing 20 wt% GO. The rapid assembly of NPSs suppresses the Plateau-Rayleigh instabilities and stabilizes the aqueous jet of Ti₃C₂Tₓ/GO. **b–e** FESEM images of Ti₃C₂Tₓ/GO worm-like aerogels. The red arrow in (**d**) highlights the laminated skin that covers the bulk of aerogels. Scale bars in (**a–e**) correspond to 2 cm, 2000, 500, 200, and 100 µm, respectively.

insertion of the GO, but one-half of the non-favorable GO basal plane/water contacts are eliminated[5]. This represents a significant gain in the free energy and a reduction in the equilibrium interfacial tension (IFT) from ~49 to 30 mN/m (Fig. S5a)[5,22]. The POSS in the oil phase rapidly interacts with the hydrophilic functional groups on GO, making this exposed plane of the GO more hydrophobic, anchoring the GO/POSS, i.e., essentially sheet-like NPSs, to the interface, dramatically reducing the IFT to 3.25 mN/m. The rigid structure of POSS, a cubic inorganic silica-cage core with side chains, significantly increases the mechanical integrity of the GO/POSS assemblies, making them ideal for stabilizing a jetted aqueous thread[5,23]. In order to further evaluate the GO/POSS interfacial layers, the worm-like liquid threads of GO were freeze-dried and assessed by a field emission scanning electron microscope (FESEM). A compact and laminated skin covering the bulk of these highly porous tubes was observed in these constructs (Fig. 1b–e).

### Shaping mGO/GO magnetic liquid threads

Using a mixed suspension of GO and mGO, magnetic liquid threads could be produced in a similar manner. The jetting of mGO/GO suspension in hexane containing POSS and the response of the magnetic liquid tubules to an external magnetic field are shown in Fig. S7a, b and Video S3. The mixed aqueous suspension of mGO and GO contained 50 wt% of GO and mGO, with a total solid concentration of 10 mg/ml. We note that liquid threads could be produced with only mGO. However, upon freeze-drying, relatively poor structural stability and high volume shrinkage were observed due to the weak interfacial skin (Fig. S7c, d)[5]. Both were significantly improved in the hybrid system.

### Shaping Ti₃C₂Tₓ/GO conductive liquid threads

Ti₃C₂Tₓ was chosen as a benchmark for generating conductive liquid threads due to its high electrical conductivity and ability to be processed in water, unlike other types of conductive nanomaterials[24]. However, when pure Ti₃C₂Tₓ is jetted into the hexane/POSS domain, it breaks into small droplets, forming liquid beads (Fig. 1f, g, S8a–c, and Video S4). Unlike GO, which acts as a surfactant and rapidly assembles at the interface and provides structural stability, Ti₃C₂Tₓ is negatively charged and does not assemble at the oil/water interface initially[17]. At a Ti₃C₂Tₓ concentration of 10 mg/ml and without POSS in the surrounding hexane phase, the equilibrium IFT is ~48 mN/m, almost equal to that of pure water against hexane, which is ~49 mN/m (Fig. S5a). In this case, the POSS must first assemble at the interface, followed by diffusion of the Ti₃C₂Tₓ that interacts with the assembled POSS[17,18,25,26].

While POSS significantly reduces the interfacial energy, the binding energy per POSS is not sufficient to withstand the compressive forces exerted when droplets begin to form. Consequently, the stream breaks up into droplets[5,13].

A solution of POSS in hexane against pure water has an IFT of ~18 mN/m, demonstrating the surfactant-like characteristics of POSS[1,13]. However, with POSS being dissolved in hexane, the equilibrium IFT of 10 mg/ml Ti₃C₂Tₓ in water against hexane containing POSS dramatically decreased to ~10 mN/m (Fig. S5c, d), confirming the formation and assembly of the Ti₃C₂Tₓ/POSS NPSs at the interface[5,23,27]. Evidence of the interfacial Ti₃C₂Tₓ/POSS assembly is also found in the FESEM images of freeze-dried Ti₃C₂Tₓ aerogel beads, where a compact, layered skin, distinct from the porous interior of the cells forming the aerogel, is evident. This indicates the formation of NPS assemblies, even when using pure Ti₃C₂Tₓ (Figs. 1h–j and S8d–h). We note that pure Ti₃C₂Tₓ also assembles at the interface, though the assemblies are much weaker in comparison to those formed with GO, which has an equilibrium IFT of ~3.25 mN/m against a solution of POSS in hexane (Fig. S5b).

To address the stability of MXene jetted streams, a mixture of GO and Ti₃C₂Tₓ was used that yielded aqueous tubules similar to that found with GO, where the functionality of the final constructs, i.e., electrical conductivity, was attained from Ti₃C₂Tₓ and the interfacial NPSs assemblies mainly relied on the GO's interfacial activity (Figs. 2a and S9 and Video S5). The equilibrium IFT of aqueous Ti₃C₂Tₓ/GO suspensions against pure hexane was considerably reduced, when compared to pure Ti₃C₂Tₓ suspension, demonstrating the superior interfacial activity of GO (Fig. S5a). This reduction was even more evident when POSS was added to the hexane, where, for suspensions of 20 and 50 wt% GO, i.e., 2 and 5 mg/ml of GO in 10 mg/ml Ti₃C₂Tₓ/GO suspensions, the IFTs were ~2.75 and 2.5 mN/m, respectively (Fig. S5b–d). These results confirm that the GO and POSS strongly assemble at the interface and stabilize the Ti₃C₂Tₓ/GO streams. The formation of interfacial Ti₃C₂Tₓ/GO/POSS assemblies is also evident in FESEM images of jetted Ti₃C₂Tₓ/GO tubules, where a laminated skin covers the jetted stream of these highly porous tubes (Figs. 2b–e and S10).

Figures S11–S14 also demonstrate the potential for precise adjustment of the dimensions of aqueous tubes and the macro-scale voids between Ti₃C₂Tₓ/GO tubules through a liquid streaming approach. As illustrated in Figs. S11 and S12, the size of the extruded tubules can be varied from 450 to 850 µm by only using needles with

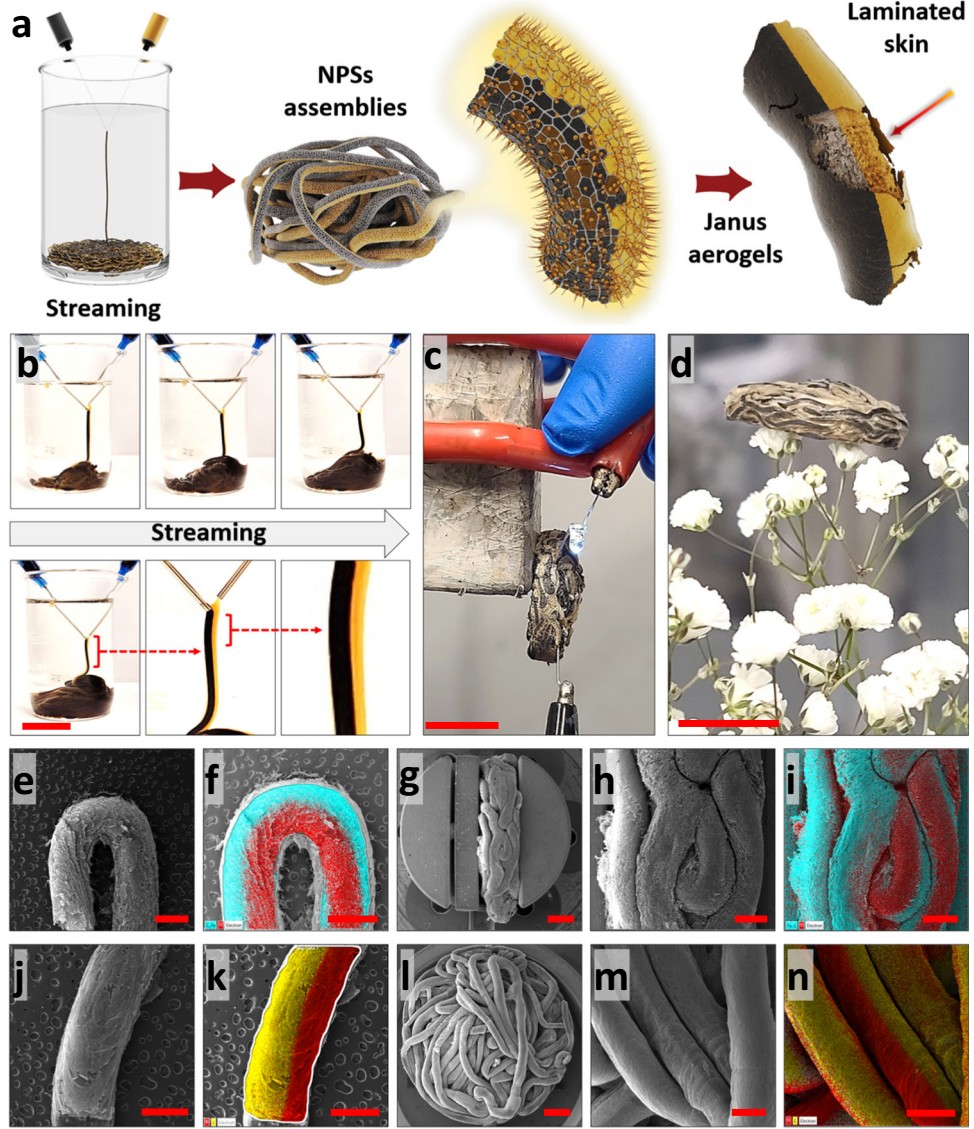

**Fig. 3 | Shaping Janus liquid threads. a** Schematic illustration of Janus liquid threads formation. In this graphical representation, GO and Ti$_3$C$_2$T$_x$ nanosheets are represented by gold and black colors, respectively. **b** Magnetic/conductive Janus structures were fabricated by merging streams of Ti$_3$C$_2$T$_x$/GO (20 wt% GO) and mGO/GO (50 wt% GO). **c, d** Digital images of Ti$_3$C$_2$T$_x$/GO:mGO/GO magnetic/conductive aerogels. FESEM and EDX analysis of (**e–i**) Ti$_3$C$_2$T$_x$/GO:mGO/GO and **j–n** Ti$_3$C$_2$T$_x$/GO:GO Janus aerogels. Scale bars in (**b–d**) and (**e–n**) correspond to 2 cm and 1000 μm, respectively.

different gauge numbers ranging from 21 to 15. Additionally, the presence of macro-scale voids among the Ti$_3$C$_2$T$_x$/GO tubules is clearly evident in the cross-sectional X-ray micro-computed tomography (micro-CT) images of these structures (Figs. S13 and S14). The size and distribution of these macro-scale pores can be regulated by altering the dimensions of the extruded filaments. This remarkable flexibility in controlling the macro-scale porosity of Ti$_3$C$_2$T$_x$/GO worm-like structures opens up exciting possibilities for diverse applications, including electromagnetic interference (EMI) shielding, oil absorption, and gas absorption/detection, where precise control over macro-to-micro scale porosities is of paramount importance.

### Fabrication of Janus liquids and aerogels

Magnetic/conductive and conductive/non-responsive Janus liquid tubules were fabricated by merging aqueous streams of 10 mg/ml Ti$_3$C$_2$T$_x$/GO (20 wt% GO) and mGO/GO (50 wt% GO) or pure GO (Fig. 3a, b and S15 and Videos S6 and S7). The heterogeneous content of the opposing sections of the Janus threads is shown in Fig. S16a–d. Interestingly, the ratio of magnetic and conductive or non-responsive parts

of the Janus liquid tubules can be readily controlled by tuning the jetting parameters. For example, using needles with different inner diameters for jetting each suspension, e.g., 200 μm for Ti$_3$C$_2$T$_x$/GO and 400 μm for mGO/GO, enabled the fabrication of Janus liquid tubules where the volume of the magnetic portion was larger than that of the Ti$_3$C$_2$T$_x$ suspension.

The Janus structured liquids were transformed to lightweight magnetic/conductive aerogels, e.g., $\rho \sim 7$ mg/cm$^3$ in the case of Ti$_3$C$_2$T$_x$/GO:mGO/GO structures printed with 200 μm needles (Fig. 3c, d and Video S8). The electrical conductivity of the Janus aerogels arises from the Ti$_3$C$_2$T$_x$-containing sections, while the magnetic response is attributed to mGO present on the opposite side of the filaments. Compared to conventional magnetic/conductive composite aerogels, the functionalities of Janus aerogels arise from different macroscopic sections of the constructs and can be easily tuned independently, i.e., alternating layers of magnetic and conductive NPs. In this way, we addressed the current challenge of electrically conductive network rupture upon the introduction of less-conductive magnetic particles[28,29].

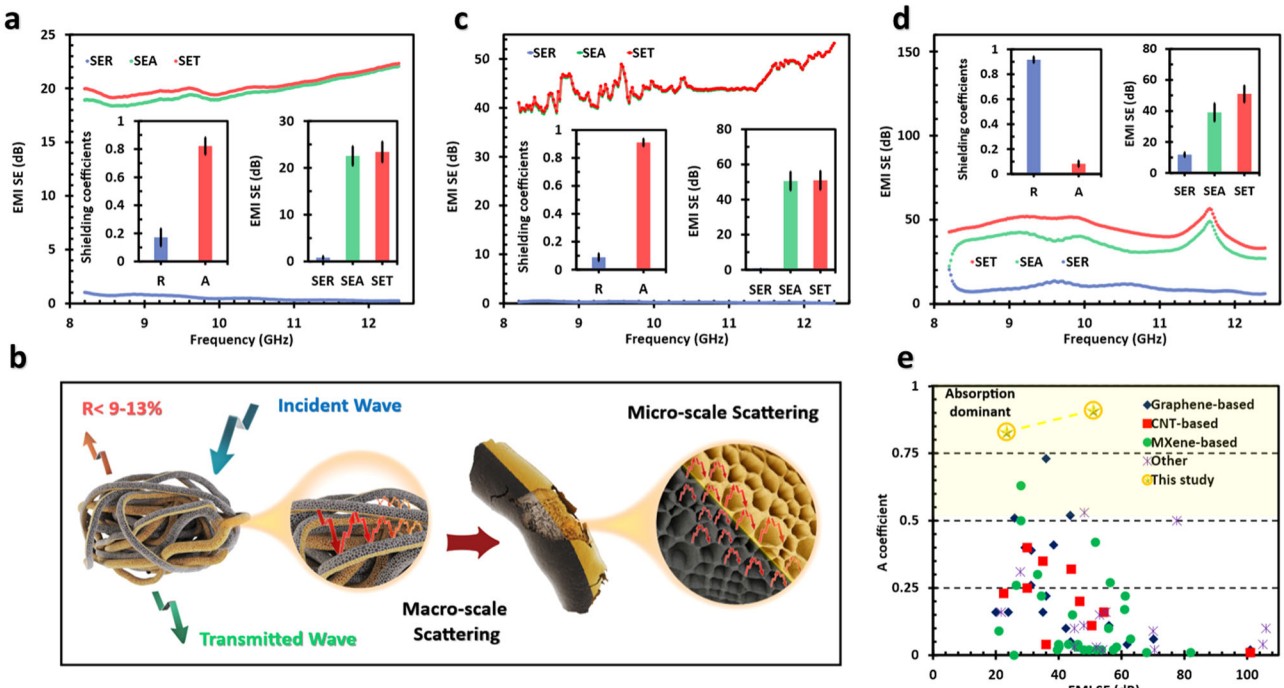

**Fig. 4 | Janus aerogels for EMI shielding. a** EMI shielding characteristics of $Ti_3C_2T_x$/GO (20 wt% GO) and mGO/GO (50 wt% GO) Janus aerogels prepared by two 200 μm nozzles. **b** Schematic illustration of internal scattering in $Ti_3C_2T_x$/GO (20 wt% GO) and mGO/GO (50 wt% GO) Janus aerogels from macro-to micro-scale. **c, d** EMI shielding characteristics of copper coated magnetic/conductive Janus aerogels. This shield consists of a 3 mm Janus aerogel covered with 0.05 mm conductive copper tape from one side. The Janus aerogel of this shield is designed in a way that the magnetic parts' volume is larger than the conductive domain. **c** summarizes the EMI shielding characteristics of the aerogel sides, while **d** showcases those characteristics that belong to the copper side of the shield. **e** A comparison of the $SE_T$ vs. $A$ of the magnetic/conductive Janus structures with the previous best practices in the literature. For more information, see Table S1.

The morphology of the Janus structures was investigated by FESEM and energy-dispersive X-ray spectroscopy (EDX) analysis and is shown in Figs. 3e–n, S16e–l, and S17–S21. The Janus aerogels show dual porosities, which is the characteristic of worm-like aerogels, i.e., macro-scale porosities between the Janus tubes and micro-scale porosities within the tubules. Figures S17 and S18 illustrate the controllable manipulation of tube size and the subsequent connections and entanglements between tubules in Janus structures achieved by employing needles of different sizes during the streaming process. Additionally, to emphasize the micro-scale porosities within these aerogels, Figs. S19 and S20 display FESEM images of hierarchical aerogels fabricated from pure GO ink, mGO/GO (20 wt% GO), and $Ti_3C_2T_x$/GO (20 wt% GO) suspensions. It's noteworthy that the formation of these micro-scale pores results from the crystallization of ice during the freezing process, which is subsequently preserved upon freeze-drying. Furthermore, the heterogeneous distribution of NPs throughout the $Ti_3C_2T_x$/GO:mGO/GO and $Ti_3C_2T_x$/GO:GO tubules is evident in EDX analysis, where a distinct boundary on the surface of the Janus aerogels filament is observed (Figs. 3e–n and S21).

The interfacial skin of these Janus worm-like constructs plays a crucial role in determining their mechanical stability[5]. In comparison to worm-like aerogels of $Ti_3C_2T_x$/GO, both Janus aerogels of $Ti_3C_2T_x$/GO:GO and $Ti_3C_2T_x$:mGO/GO showed enhanced compressibility and mechanical stability (Fig. S22 and Videos S9–S12). These characteristics can be explained by the formation of a GO network throughout the tubes and the altered stress distribution on $Ti_3C_2T_x$/GO parts that have a semi-cylindrical shape in the case of Janus tubes[5,30].

## Janus EMI shields with absorption-dominant characteristics

Before discussing the absorption-dominant behavior of Janus aerogels, an examination of the EMI shielding characteristics of the worm-like structures composed of $Ti_3C_2T_x$/GO (20 wt% GO), mGO/GO (50 wt% GO), and pure GO is conducted. It is noteworthy that both mGO/GO (50 wt% GO) and pure GO fail to exhibit the impedance mismatch characteristics crucial for effective EMI shielding, primarily due to their lack of electrical conductivity. Consequently, these structures are considered unsuitable for mitigating unwanted electromagnetic interference, as exemplified in Figs. S23 and S25.

In contrast, the conductive worm-like structures of $Ti_3C_2T_x$/GO (20 wt% GO) demonstrate an exceptional shielding effectiveness of 69.2 dB, a noteworthy achievement for lightweight electromagnetic shields (Fig. S23). Importantly, this EMI shielding behavior is achieved without the need for thermal or chemical reduction processes. However, it is crucial to emphasize that the conductive properties of these structures, characterized by an electrical conductivity of approximately $120 ± 10$ S/m, result in a high reflectance index of approximately 70%, corresponding to an absorbance of around 30%. This high reflectance introduces secondary reflection issues, a challenge that we endeavor to address through the development of Janus aerogels.

To underscore the future potential of the Janus liquid constructs, magnetic/conductive Janus aerogels were used to address a key challenge in EMI shielding: the high reflection index. The Janus aerogels of $Ti_3C_2T_x$/GO (20 wt% GO) and mGO/GO (50 wt% GO), jetted with two 200 μm nozzles, showed exceptional absorption-dominant shielding behavior, e.g., $SE_T = 23.4$ dB, $SE_R = 0.8$ dB, and $A = 83%$ (Figs. 4a and S26). This unique behavior can be explained by the high porosity ranging from the micro- to macro-scales and alternating magnetic/conductive domains of the aerogels, creating numerous interfaces within a small volume. Here, by incorporating a magnetic component (mGO/GO), Janus aerogels ensure magnetic loss, while the conductive element ($Ti_3C_2T_x$/GO) contributes to dielectric loss, enabling efficient absorption and dissipation of electromagnetic waves[31,32]. Due to the

very low electrical conductivity of magnetic domains, their impedance almost matches free space and allows the penetration of EM waves into the bulk of aerogels[33]. Even with conductive domains, a large portion of EM waves can pass through due to the shallow thickness of conductive sections that always have magnetic domains next to them[34,35]. Upon penetrating these aerogels, the energy of EM waves gets dissipated due to numerous interfaces arranged inside these structures from micro- to macro-scales and prolonged the interaction of the waves with electrical/magnetic dipoles and domains with finite electrical conductivity. In addition, the Janus structures have numerous interfaces between conductive and magnetic domains with impedance match/mismatch characteristics that can further boost the scattering and absorption of EM waves. This unique absorption mechanism, schematically represented in Fig. 4b, not only safeguards human health and sensitive equipment from EM waves but also minimizes the secondary EMI pollution, which is as detrimental as the original waves.

Using the magnetic/conductive Janus aerogels, we also developed a unique EMI trap with $SE_T = 51$ dB, $SE_R = 0.4$ dB, and $A = 91\%$. This structure is comprised of a 3 mm Janus aerogel covered with 0.05 mm conductive copper tape from one side. From the aerogel sides, we observed a unique absorption behavior, while the copper side mainly reflected the EM waves, i.e., similar to that of pure copper tape (Figs. 4c, d and S27–S29). The Janus aerogel was prepared so that the magnetic section's volume fraction was larger than that of the conductive section; specifically, mGO/GO (50 wt% GO) and $Ti_3C_2T_x$/GO (20 wt% GO) suspensions were jetted from 400 and 200 µm nozzles, respectively. Therefore, the impedance mismatch between this Janus structure and free space was even lower than the Janus aerogels described previously, allowing the incident waves to penetrate the structure with minimum reflection. Once the EM waves penetrate the structure, their energy gets dissipated due to the presence of multi-scale porosities and alternating magnetic/conductive layers that significantly boost the internal scattering. This construct is essentially a trap for EM waves. Figure 4e and Table S1 compare the $SE_T$ and $A$ of this shield with others. The exceptional EMI shielding performance and unique absorption-dominant behavior make these structures ideal for medical and military applications, where $SE_T > 30$ dB and a low reflection index are of utmost importance[34,36].

## Janus piezoresistive sensors for human motion monitoring

The Janus aerogels of $Ti_3C_2T_x$/GO (20 wt% GO) and GO show excellent reversible compressive behavior, where the structures that were compressed up to 70% rapidly recovered to their original shape after release (Fig. 5a, b). The stress-strain curves of these Janus aerogels at different sets of strains show mechanical robustness, which is crucial for high-performance piezoresistive sensors (Fig. 5a). Fatigue hysteresis tests were performed on the prepared aerogels, subjecting them to 100 compressive cycles at large strains of 40% and 50%. The results showed minor plastic deformation, where the aerogels maintained ~80% of their initial Young's moduli and maximum stresses (Fig. S30). Besides, the interfacial skin that covers the bulk of the aerogels maintains the mechanical integrity of these structures that remained intact after 100 compressive cycles, indicating their mechanical robustness (Fig. S31).

The cyclic piezoresistive sensing performance of the Janus aerogel under different compressive strains is shown in Fig. 5c. Upon compression, the density of the interconnected conductive sections of the Janus aerogels on micro- and macro-scales increases, causing an exponential increase in conductance. By increasing the compressive strain to 60%, face-to-face contact between adjacent conductive domains becomes more effective, leading to a higher sensitivity. Additionally, it is important to mention that the inclusion of GO on the opposite side of the Janus aerogels was aimed at improving the overall mechanical properties and compressibility, which cannot be achieved solely with the worm-like $Ti_3C_2T_x$/GO (20 wt% GO), as depicted in Fig. S22 and Videos S9–S12, along with their corresponding discussion.

The performance of pressure sensors is evaluated based on their sensitivity, i.e., $S = \frac{\frac{\Delta I}{I_0}}{\Delta P}$, and response/recovery times, which are calculated to be 1-8.18 kPa$^{-1}$ and 300/250 ms for the Janus aerogels, respectively (Fig. 5d–f)[37,38]. These characteristics are comparable to some of the best sensors available with the significant advantage that no chemical/thermal reduction or pre-processing, e.g., freeze-casting, is needed for the Janus aerogels (see Table S2 and the corresponding discussion). Consequently, these interfacially driven morphologies open up numerous opportunities in pressure sensing and human motion monitoring.

The Janus aerogels were also used as touch sensors, where the touch strength and frequency can be obtained by analyzing the signal curves (Fig. 5g, h and Videos S13 and S14). To assess the durability of these sensors, five participants were asked to touch the sensor, and the response was recorded over the course of the experiment (Figs. 5i, j and S32). Even after 2000 compression cycles, the baseline of the sensor and response/recovery signals were stable, underscoring their potential for applications ranging from wearable electronics to human-machine interfaces. The response of the sensors remained stable for up to three months (Fig. S33 and Video S15). Only the oxidation of $Ti_3C_2T_x$ lowered the baseline of the sensor, which is translated into a lower electrical conductivity[34,39]. However, the Janus aerogels' response/recovery signals and piezoresistive sensing performances remained stable.

The Janus aerogels of $Ti_3C_2T_x$/GO were also used for real-time human motion and health monitoring. Stable, reproducible signals were produced from the repeated bending of fingers and contraction–relaxation of biceps muscles, both large-scale body movements (Fig. 5k, l). Subtle motions, including the movement of facial muscles, talking, and coughing, were also monitored. The Janus aerogel devices could distinguish the vocalization of different letters and even polysyllabic words when attached to the skin outside of the larynx of a human subject (Figs. 5m and S34 and Video S16). The response of this system to coughing is shown in Fig. 5n, demonstrating the feasibility of our sensor for health monitoring devices. The sensitivity to facial movement is shown in Fig. 5o for multiple cycles of contraction and relaxation of forehead muscles. This response can possibly be used for human emotion recognition as one of the many potential applications of Janus aerogels.

In summary, Janus all-liquid systems with anisotropic and customizable distributions of NPs were developed through interfacial assembly and jamming of NPs at the liquid-liquid interface. These Janus liquids offer greater versatility and tailorability when compared to conventional composites, as they allow for the precise, independent tuning of different functionalities on the different sides of the Janus architecture, resulting in multi-responsive soft materials with well-defined characteristics. The potential of these anisotropic platforms has been demonstrated through proof-of-concept applications such as absorption-dominant EMI shielding, piezoresistive sensing, and human motion monitoring. We anticipate that the Janus liquid approach will have many more potential uses in areas such as dual-encapsulation of cells and active matter, vessels for biphasic chemical synthesis and chemical separations, multi-modal sensors, and all-liquid robots, where the simultaneous encoding of multiple functionalities in a single construct is highly beneficial. Overall, our results suggest that Janus liquids are unique alternatives to traditional composites, blends, and grafted systems, where different functionalities can be achieved simultaneously in a single structure without interference.

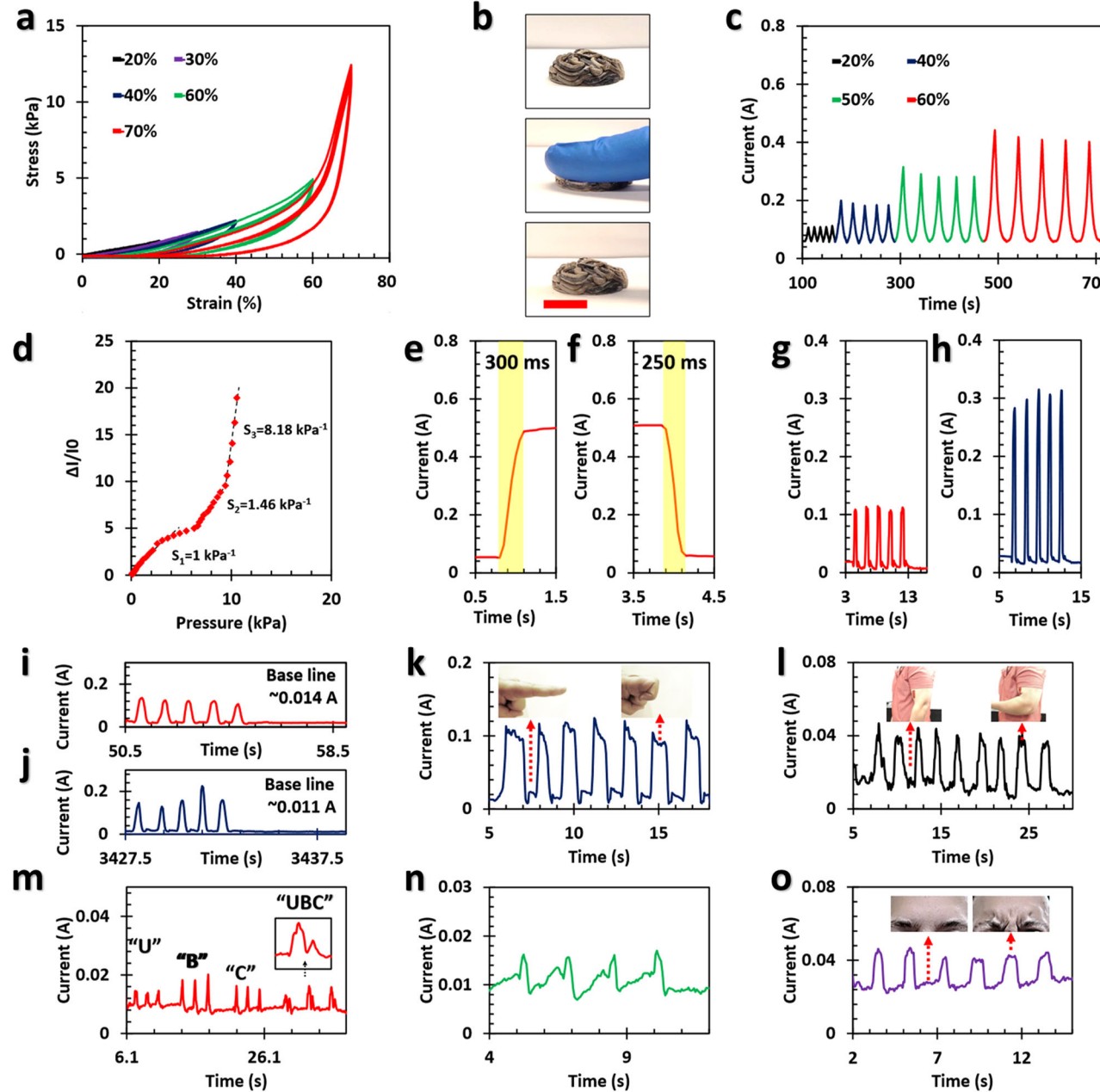

**Fig. 5 | Janus aerogels for pressure sensing and human motion monitoring.**
**a** The stress–strain curves of the Janus aerogels of $Ti_3C_2T_x$/GO (20 wt% GO):GO at different sets of strains. **b** Digital images showcasing the compressibility of $Ti_3C_2T_x$/GO (20 wt% GO):GO aerogels. **c** The cyclic piezoresistive sensing performance of the Janus aerogel under different compression strains. The **d** sensitivity and **e**, **f** response/recovery times of the fabricated sensor. The recorded signal for **g** soft and **h** hard press of the sensor. The recorded response of the sensor **i** before and **j** after 2000 bending cycles under real conditions. The capability of the sensor for monitoring **k** finger movements and **l** contraction–relaxation biceps muscles, both as large-scale body movements. The recorded response for various subtle human motions, including **m** talking, **n** coughing, and **o** facial muscle movements. Scale bar in (**b**) corresponds to 2 cm.

## Data availability

All data generated in this study have been securely deposited on Figshare and are openly accessible at the following: https://doi.org/10.6084/m9.figshare.24455059.

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

## Acknowledgements

The authors acknowledge the Syilx Okanagan Nation for the use of their traditional, ancestral, and unceded territory, the land on which the research was conducted. A.G., S.A.H., F.A., M.G., and M.A. acknowledge the financial support of the Natural Sciences and Engineering Research Council of Canada (NSERC), funding reference number ALLRP 555586-20, and Zentek Ltd. M.A. appreciates the financial support provided by the Canada Research Chairs Program (CRC-2018-00234). F.A., O.J.R., and S.E.M. are grateful for funding support provided by the Canada Excellence Research Chair Program (CERC-2018-00006) and the Canada Foundation for Innovation (Project number 38623). M.S. expresses gratitude for the financial assistance provided by Grant No. CMMI-2134607 from the U.S. National Science Foundation. M.K. acknowledges the support of the NSERC Discovery Grant, with funding reference number RGPIN-2023-03466. T.P.R. was supported by the U.S. Department of Energy, Office of Science, Office of Basic Energy Sciences, Materials Sciences and Engineering Division under Contract No. DE-AC02-05-CH11231 within the Adaptive Interfacial Assemblies Towards Structuring Liquids program (KCTR16). O.Z. and S.W. acknowledge the support of the Spanish State Research Agency (AEI) and the European Regional Development Fund (ERDF) under project PID2020–15935RB-C42. All authors express deep appreciation to Fipke Laboratory for Trace Element Research (FiLTER), located within the Irving K. Barber Faculty of Science at the University of British Columbia, for their invaluable assistance with FESEM analysis in this paper.

## Author contributions

A.G., S.A.H., T.P.R., M.K., and M.A. conceived the project. A.G. and S.A.H. conducted the experimental work and wrote the manuscript. S.E.M. performed interfacial tension measurements and participated in discussions. F.A.J., M.G., H.R., O.Z., and M.S. assisted with nanomaterial synthesis and characterizations. O.J.R., S.W., T.P.R., M.K., and M.A. oversaw the project. All authors discussed the results and made contributions to the manuscript.

## Competing interests

The authors declare no competing interests.
