## [Peer Review File · Nature Communications]

REVIEWER COMMENTS

Reviewer #1 (Remarks to the Author):

The authors have proposed the concept of “Janus liquids” in the manuscript. This concept, although originates from the microfluidic technology, the authors combined it with nanoparticle interfacial assembly, and obtained materials exhibiting unique structure and properties. They also show some potential applications. Overall, the reviewer believes the idea is original and seems interesting, but some of the issues should be addressed before further consideration.

1. More characterizations and tests should be done with regard to the mechanical properties of the resultant aerogels.
2. The explanation on the advantages and enhanced performance of “Janus” is too simple. Why Janus is important in those applications? The authors should make more and deeper explanation.
3. How to realize controllable processing of the bulk tubules made from Janus liquids?
4. How nanoparticles function in applications including piezoresistive sensing, human motion monitoring, and electromagnetic interference shielding? Are there any changes of the nanoparticles, especially in microscopic scale? The relationships between building blocks and applications should be made.

Reviewer #2 (Remarks to the Author):

This manuscript studies the formation of multifunctional aerogels fabricated using multi streams of aqueous NP dispersions into the oil. The manuscript is well-written, but there are some concerns about the focus of the papers described below.

1. The paper starts with an introduction about structured liquids and all liquids systems and how to create liquid threads, but for the rest of the paper, the focus of this work heavily relies on the functionality of aerogels. The title and emphasis of the work are more focused on the functional Janus liquids or structured liquids, which is not descriptive of the full studies. If the work can be restructured to frame how to achieve new aerogels using multi-liquid-liquid techniques, it can make it more focused for the readers. It is also interesting if the authors include what improvement in the

system can be made (e.g., 3D printing, coaxial needles). The authors should provide a bigger picture of how this work is related to other liquid-liquid systems in the fields.

2. Using Janus term in the title and the text for structure and fluids is misleading and not descriptive of the system; perhaps the term multi or combined liquid streams can better describe the new system studied in this work. Since Janus particles are a defined term in materials science, using the term Janus for liquid/structure can be confusing. The same issue to the term "liquid streaming"; it is uncommon. It is recommended to use simpler, more common terminologies that are more descriptive of the systems.

3. The use of programming for this system is overstated. Programming has different meanings in the field, and quantitative programming is not indicated in the current study.

4. Page 13, line 225, it is not clear how the functionality of the systems is impacted by multi-scale porosity.

Reviewer #3 (Remarks to the Author):

Manuscript #: NCOMMS-23-14122-T

Recommendation: Reject

(Lacks novelty, authors have already published similar works in multiple journals, not suitable for Nature Communications)

The following manuscript entitled "Functional Janus liquids" presents a study of preparing conductive lightweight hybrid of magnetic graphene oxide (mGO) and Ti₃C₂T_x MXene with Janus structure. The present work is interesting with very low reflection of EM waves; however, the authors have already published similar works in different journals (1. Small 2022, 18, 2200220; and 2. Advanced Materials Interfaces 2022, 9, 2101659, etc.). The present work is a total reproduction (with the same presentation of figures and results) of the works published by the authors before. In previous publications, authors used graphene oxide in liquid streaming to get interfacial skin of GO in POSS-incorporated nonpolar (e.g., Hexane) solvent, whereas the current study includes MXene with the GO to obtain the same structure. Authors didn't provide any comparison of their own studies, that what makes a difference between GO/POSS and GO-MXene/POSS systems. Furthermore, the current manuscript lacks a sufficient understanding and explanation of the mechanisms behind the absorption characteristics of EMI shielding. Therefore, due to lack of novelty

of the idea, this paper can't be published in prestigious journal of Nature Communications, and hence rejected.

Reviewer #1 (Remarks to the Author):

The authors have proposed the concept of “Janus liquids” in the manuscript. This concept, although originates from the microfluidic technology, the authors combined it with nanoparticle interfacial assembly, and obtained materials exhibiting unique structure and properties. They also show some potential applications. Overall, the reviewer believes the idea is original and seems interesting, but some of the issues should be addressed before further consideration.

We thank the Reviewer for the comments and insightful feedback on our paper, which helped improve the readability and impact of our manuscript. We provide a point-by-point response to each of the referee’s comments.

1. More characterizations and tests should be done with regard to the mechanical properties of the resultant aerogels.

Response: We conducted fatigue hysteresis tests on $\text{Ti}_3\text{C}_2\text{T}_x/\text{GO}$ (20 wt% GO) and GO Janus aerogels at two different strain levels (40 and 50%) for up to 100 compression cycles. The results showed that the samples had only minor plastic deformation under the testing conditions; Young's modulus retention of ~ 80% of the initial value was observed for aerogels at the maximum stress. To understand the mechanical stability of the samples, which primarily arises from the interfacial layer, we performed SEM imaging on $\text{Ti}_3\text{C}_2\text{T}_x/\text{GO}$ (20 wt% GO) and GO Janus aerogels after fatigue (following 100 compression cycles): the interfacial skin covering the bulk of the aerogels remained intact, a factor that favored the mechanical integrity (Figure S18).

We also note that we have additional characterizations to assess the mechanical performance of the prepared samples. These include:

- SEM analysis indicating the presence of layered skins in the aerogels (Figures 3 and S12).
- Stress-strain profiles for the Janus aerogels at different strains (see Figure 5a).
- Evaluation of the sensor's durability after 2,000 compression cycles under real conditions. This analysis is of significance (Figures 5i, 5j, and S19), since it confirmed the response of the sensors, which remained unchanged. The observation is attributed to the mechanical stability of the Janus aerogels. We did not observe any noticeable plastic deformation, which would have led to sensing variations.

The additional characterizations further contribute to our understanding of the mechanical properties and performance of the Janus aerogels and provide valuable insight. The supplementary information is now revised with the new data, Figures S17 and S18, in the revised document. Also, we have incorporated the following sections in the main manuscript:

The Janus aerogels of $\text{Ti}_3\text{C}_2\text{T}_x/\text{GO}$ (20 wt% GO) and GO show excellent reversible compressive behavior, where the structures that were compressed up to 70% rapidly recovered to their original shape after release (Figure 5a-b). The stress-strain curves of these Janus aerogels at different sets of strains show mechanical robustness, which is crucial for high-performance piezoresistive sensors (Figure 5a). Fatigue hysteresis tests were performed on the prepared aerogels, subjecting them to 100 compressive cycles at large strains of 40% and 50%. The results showed minor plastic deformation, where the aerogels maintained ~80% of their initial Young's moduli and maximum stresses (Figure S17). Besides, the interfacial skin that covers the bulk of the aerogels maintains the mechanical integrity of these structures that remained intact after 100 compressive cycles, indicating their mechanical robustness (Figure S18).

In supplementary data, we have:

Figure S17: Fatigue hysteresis tests of $\text{Ti}_3\text{C}_2\text{T}_x/\text{GO}$ (20 wt% GO) and GO Janus aerogels. The samples were compressed up to (a) 40% and (b) 50% strain for 100 cycles. The results showed minor plastic deformation at those strain levels, where the aerogels maintained ~80% of their initial Young's moduli and maximum stresses.

Figure S18: SEM images of Janus aerogels of $\text{Ti}_3\text{C}_2\text{T}_x/\text{GO}$ (20 wt% GO) and GO after 100 compression cycles. The sample shown in (c) was compressed during SEM imaging. All scale bars in (a-c) correspond to 1 mm.

2. The explanation of the advantages and enhanced performance of “Janus” is too simple. Why Janus is important in those applications? The authors should make more and deeper explanations.

The reviewer raises an important point, and therefore we have incorporated a section in the supporting information and revised the last two paragraphs of the introduction to highlight the following points:

- Significance of Janus liquids and aerogels: We describe why Janus liquids and aerogels are of considerable importance in various applications.
- Assigning different functionalities to different parts of Janus liquids: We expand on the methodology used to assign distinct functionalities to specific regions of the Janus liquids.
- Significance of Janus liquids as templates for fabricating aerogels with simultaneous function control: We emphasize the crucial role played by Janus liquids as liquid templates in the fabrication of aerogels.

Specifically, the following text is now added to the revised manuscript:

Here, we report a one-step fabrication of Janus structured liquids with an anisotropic distribution of NPs. Our approach relies on stabilizing and then merging two aqueous streams of NPs in a nonpolar liquid containing complementary functionalized ligands. The resulting liquid threads consist of two distinct sections, each composed of specific groups of NPs. This unique duality inherent in Janus liquids affords an opportunity to assign distinct functionalities to opposing sides of the structures and fine-tune them independently. As a proof of concept and to highlight the

future potential of these unique constructs, magnetic, conductive, or non-responsive NPs were sequestered in opposite sides of these constructs using magnetic graphene oxide (mGO)/GO, $\text{Ti}_3\text{C}_2\text{T}_x/\text{GO}$, or GO suspensions, allowing the formation of multi-responsive soft materials that can be heterogeneously tailored from the micro- to macro-scale.

Such precise control over the functionality of Janus constructs becomes even more pronounced when these structured liquids are used as templates, enabling the development of customized compositions and arrangements for task-oriented aerogels. The potential applications of Janus aerogels are vast, especially when a deliberate connection is established between the functionality of the Janus building blocks and the ultimate application. For instance, the development of Janus aerogels with non-interfering magnetic/conductive opposing sections represents a breakthrough in electromagnetic interference (EMI) shielding. By integrating alternating magnetic/conductive domains within the aerogel structure, Janus constructs surpass existing shielding materials, offering superior wave absorption properties. This addresses a long-standing challenge in EMI shielding and positions Janus aerogels as highly promising candidates for next-generation shielding materials. This integration platform is also used to make functional aerogels flexible, where functionality, e.g., electrical conductivity, is customized through one side of the structures, and mechanical flexibility is mainly derived from the opposite side. This adjustable strategy opens numerous opportunities in pressure sensing and human motion monitoring.

The revised supporting information includes new text as follows:

Rationalization of Janus design: Controllable functionality and liquid templating concept

Janus liquids can be visualized as liquid threads consisting of two distinct sections, i.e., faces, each composed of a specific group of NPs. This duality embodies the essence of Janus, representing a remarkable opportunity to assign distinct functionalities to opposing sides of these structures and

independently fine-tune them. This landmark nature of the Janus liquids becomes even more pronounced when they are used as a liquid template, enabling the development of customized compositions and arrangements for task-oriented aerogels.

The possible uses of Janus aerogels are limitless, especially when a deliberate connection is established between the functionality of the Janus building blocks and the ultimate application. For instance, developing Janus aerogels with non-interfering magnetic/conductive opposing sections is valuable for EMI shielding applications where alternating magnetic/conductive domains of the aerogels create numerous interfaces within a small volume to absorb the electromagnetic waves (see the subsequent discussion in the following sections and Supplementary Information). One other application involves the development of Janus aerogels, where one section of the structure exhibits specific functionality, while the other imparts desired mechanical characteristics, as described in this paper. Overall, this anisotropic integration platform with adjustable functionality has many potential applications where the simultaneous encoding of multiple functionalities in a single construct is highly beneficial, e.g., Janus gas/liquid absorbers, where each section is fine-tuned to deal with a specific type of material, or multi-modal sensors that can accurately and independently sense two different classes of analytes.

3. How to realize controllable processing of the bulk tubules made from Janus liquids?

Response: The controlled processing of Janus liquids is a critical subject. We categorize controllable processing into two key areas: 1) the manipulation of concentration, composition, and assigned functionality of Janus liquid threads, and 2) the control of the process of liquid Janus structuring to achieve a predefined geometry by 3D printing. Our revised manuscript provides additional guidelines regarding the first area:

- Controlling the concentration and composition of the building blocks of Janus liquid threads:** Janus liquid threads with a customizable NPs distribution throughout their structures are fabricated by joining two aqueous streams of NP dispersions in an apolar medium. In this method, the concentration and composition of each building block of the Janus liquids are effectively modulated. For example, the conductive liquid threads of $\text{Ti}_3\text{C}_2\text{T}_x/\text{GO}$ with different compositions and concentrations are shown in Figures 2, S9, and S10. This wide range of options for the conductive portion of Janus structured liquids allowed us to control the composition of the conductive part, thereby influencing the density and electrical conductivity of the subsequent section in Janus aerogels. The same flexibility is also observed for the non-responsive portion of the fabricated structures, where various suspensions of GO with concentrations ranging from 5 to 10 mg/ml were developed and used in liquid jetting (Figures 1a-e and S6).
- Controlling the relative ratio of the opposing building blocks of Janus liquid threads:** The ratio of opposing parts in Janus liquid threads can be controlled by adjusting the jetting parameters. For example, employing different needles for each ink (such as 200- and 400- μm needles for $\text{Ti}_3\text{C}_2\text{T}_x/\text{GO}$ and mGO/GO, respectively, allowed the successful fabrication of Janus structures with a larger volume of the magnetic region compared to the one containing $\text{Ti}_3\text{C}_2\text{T}_x$ (Figure 12 a-d).
- Controlling the functionality of Janus constructs:** The concept of Janus liquid threads offers an effective approach to assign nanomaterials with distinct characteristics to the opposite sides, thus enabling specific functionalities. In the context of this research, magnetic, non-responsive, or electrically conductive materials can be sequestered to different regions of the Janus constructs, as shown in Figures 3 and S11-13. This concept

holds great potential for numerous applications. For instance, it can be used for the development of multi-responsive gas sensors by assigning materials sensitive to different molecules to different sides of Janus structures. Furthermore, Janus structures can be designed to effectively absorb various pollutants by assigning respective nanomaterials to opposite sides of Janus constructs.

The use of Janus structured liquids for achieving predefined 3D geometries by 3D printing still requires further advancements. It is important to acknowledge that the primary challenges in this field are associated with the availability of 3D printers. To achieve the fabrication of a 3D-printed Janus structure with controlled processing of bulk Janus threads, it is essential to use two injection nozzles that remain spatially fixed relative to each other during the printing process. As the nozzles begin to move, they should maintain a predetermined separation distance. Therefore, it is necessary to design special holders for Janus printing that can be connected to the printing head. These holders should be adjustable, allowing modification of the relative needle positions as needed. Although this is easy to realize, such a setup is currently not available commercially. We are in the process of developing holders and preparing a customized printer capable of efficiently processing Janus liquids.

Additionally, based on commercial 3D printers, we connected the syringes that hold the suspension to the injection head inside a hexane bath using polymer-based tubes. These tubes move easily due to vibrations. The smallest vibration caused the movement of the injection head, which destroyed the fabricated structures. In the case of worm-like structures reported in this study, vibration creates no problem because both nozzles are stationary; therefore, we can fix the nozzles with clamps. However, for the case of 3D printing and controllable processing, we need metal tubing for fabricating Janus liquids. To address this issue, we are currently working on developing a custom

liquid printer specifically designed for fabricating Janus printed structures. We aim to thoroughly explore this topic in our future research endeavors.

The following information regarding the controllability of Janus liquids is added to the experimental section of the article:

1.2.5. Controllability of Janus Liquids: Tuning Composition and Functionality for Tailored

Constructs

Controllability in the fabrication of Janus liquids allows the composition and functionality of the final constructs to be independently tuned. This controllability can be defined through the following means:

1. Control over concentration and composition: Janus liquid threads are created by joining two aqueous streams of nanoparticle (NP) dispersions in an apolar domain. This method allows effective tuning of the concentration and composition of each building block. For example, Figures 2, S9, and S10 demonstrate various conductive liquid threads of $\text{Ti}_3\text{C}_2\text{T}_x/\text{GO}$ with different compositions and concentrations. The wide range of options for the conductive portion of the Janus structured liquids enables precise control over the composition, influencing the density and electrical conductivity of the subsequent sections in Janus aerogels. The same flexibility applies to the non-responsive portion, where different suspensions of GO with concentrations ranging from 5 to 10 mg/ml were used in liquid streaming, as illustrated in Figures 1a-e and S6.
2. Control over the relative ratio of opposing building blocks: By adjusting the streaming parameters, the ratio of opposing parts of the Janus liquid threads can be controlled. For instance, using different needles for each ink, such as a 200 μm needle for $\text{Ti}_3\text{C}_2\text{T}_x/\text{GO}$ and

a 400 μm needle for mGO/GO, allows the fabrication of Janus structures with a larger volume of magnetic parts compared to the volume containing $\text{Ti}_3\text{C}_2\text{T}_x$, as shown in Figure 12 a-d.

3. Control over the subsequent functionality: The concept of Janus liquid threads enables the allocation of specific nanomaterials with distinct characteristics to opposite sides, resulting in specific functionalities. In this research, materials with magnetic, non-responsive, or electrically conductive properties were assigned to different regions of Janus constructs, as shown in Figures 3 and S11-13. This concept holds great promise for various applications, such as developing multi-responsive gas sensors or designing Janus structures for the efficient absorption of different classes of pollutants.

4. How nanoparticles function in applications including piezoresistive sensing, human motion monitoring, and electromagnetic interference shielding? Are there any changes of the nanoparticles, especially in microscopic scale? The relationships between building blocks and applications should be made.

Response: In the context of EMI shielding, the utilization of magnetic and conductive NPs on opposite sides of Janus structures creates impedance match/mismatch interfaces within a compact volume, from micro- to macro-scale, resulting in effective scattering of electromagnetic waves. Furthermore, the incorporation of a magnetic component (mGO/GO) in the Janus aerogels enables magnetic loss, while the inclusion of the conductive element ($\text{Ti}_3\text{C}_2\text{T}_x/\text{GO}$) contributes to dielectric losses, facilitating efficient absorption and dissipation of electromagnetic waves. We address this important point in the following section of the main article:

To underscore the future potential of the Janus liquid constructs, magnetic/conductive Janus aerogels were used to address a key challenge in EMI shielding: the high reflection index. The Janus aerogels of $\text{Ti}_3\text{C}_2\text{T}_x/\text{GO}$ (20 wt% GO) and mGO/GO (50 wt% GO), jetted with two 200 μm nozzles, showed exceptional absorption-dominant shielding behavior, e.g., $\text{SE}_T=23.4$ dB, $\text{SE}_R=0.8$ dB, and $A=83\%$ (Figures 4a, S15). This unique behavior can be explained by the high porosity ranging from the micro- to macro-scales and alternating magnetic/conductive domains of the aerogels, creating numerous interfaces within a small volume. Here, by incorporating a magnetic component (mGO/GO), Janus aerogels ensure magnetic loss, while the conductive element ($\text{Ti}_3\text{C}_2\text{T}_x/\text{GO}$) contributes to dielectric loss, enabling efficient absorption and dissipation of electromagnetic waves^{31,32}. Due to the very low electrical conductivity of magnetic domains, their impedance almost matches free space and allows the penetration of EM waves into the bulk of aerogels³³. Even with conductive domains, a large portion of EM can pass through due to the shallow thickness of conductive sections that always have magnetic domains next to them^{34,35}. Upon penetrating these aerogels, the energy of electromagnetic waves gets dissipated due to numerous interfaces arranged inside these structures from micro- to macro-scales and prolonged the interaction of the waves with electrical/magnetic dipoles and domains with finite electrical conductivity. In addition, the Janus structures have numerous interfaces between conductive and magnetic domains with impedance match/mismatch characteristics that can further boost the scattering and absorption of EM waves. This unique absorption mechanism, schematically represented in Figure 4b, not only safeguards human health and sensitive equipment from EM waves but also minimizes the secondary EMI pollution, which is as detrimental as the original waves.

Throughout the revised article, we have extensively discussed the significance of NPs in non-responsive/conductive aerogels. The conductive sections of Janus aerogels undergo deformation under load, leading to variations in electrical conductivity, which serves as the primary sensing mechanism in Janus sensors. Simultaneously, the incorporation of non-responsive GO nanosheets acts as reinforcement, enhancing the mechanical compressibility of the structures. For further details, please refer to Figure S14 and Videos S9-12, along with their corresponding discussion. In response to the comment, we have included the following section:

The cyclic piezoresistive sensing performance of the Janus aerogel under different compressive strains is shown in Figure 5c. Upon compression, the density of the interconnected conductive sections of the Janus aerogels on micro- and macro-scales increases, causing an exponential increase in conductance. By increasing the compressive strain to 60%, face-to-face contact between adjacent conductive domains becomes more effective, leading to a higher sensitivity. Additionally, it is important to mention that the inclusion of GO on the opposite side of the Janus aerogels was aimed at improving the overall mechanical properties and compressibility, which cannot be achieved solely with the worm-like $\text{Ti}_3\text{C}_2\text{T}_x/\text{GO}$ (20 wt% GO) as depicted in Figure S14 and Videos S9-12, along with their corresponding discussion.

Reviewer #2 (Remarks to the Author):

This manuscript studies the formation of multifunctional aerogels fabricated using multi streams of aqueous NP dispersions into the oil. The manuscript is well-written, but there are some concerns about the focus of the papers described below.

We appreciate the reviewer's assessment. We carefully considered each of the comments and provide a detailed response below.

1. The paper starts with an introduction about structured liquids and all liquids systems and how to create liquid threads, but for the rest of the paper, the focus of this work heavily relies on the functionality of aerogels. The title and emphasis of the work are more focused on the functional Janus liquids or structured liquids, which is not descriptive of the full studies. If the work can be restructured to frame how to achieve new aerogels using multi-liquid-liquid techniques, it can make it more focused for the readers. It is also interesting if the authors include what improvement in the system can be made (e.g., 3D printing, coaxial needles). The authors should provide a bigger picture of how this work is related to other liquid-liquid systems in the fields.

Response: The reviewer raises important points which we agree are relevant to contextualize the work to make it more focused and to better balance the synthesis of the structured liquids, their functions and uses. The details in response to the points raised are divided in four parts, as follows:

1 - Structured liquids, functionality of aerogel, title and emphasis: A significant portion of our paper is dedicated to the exploration of functional aerogels derived from Janus liquids. Consequently, we have revised the title to accurately reflect the content of our study: "**Functional Janus Structured Liquids and Aerogels.**"

2 - Achieving new aerogels using multi-liquid-liquid techniques: In light of this comment and the feedback received from other reviewers, we have modified the manuscript by incorporating an additional section within the supporting information. We also revised the last two paragraphs of the introduction. These new sections aim to address three key elements:

- The significance and importance of Janus liquids: We delve into the reasons why Janus liquids and aerogels hold considerable importance for certain applications.
- Assigning different functionalities to different parts of the Janus liquids: We provide an explanation of the methodology employed to assign distinct functionalities to specific regions of Janus liquids.
- Highlight the significance of Janus liquids as templates for fabricating aerogels with controllable functionalities: We emphasize the importance and advantages of utilizing Janus liquids as templates in the fabrication of aerogels, enabling the achievement of multiple functionalities that can be controlled simultaneously.

The revised text includes the following paragraphs:

Here, we report a one-step fabrication of Janus structured liquids with an anisotropic distribution of NPs. Our approach relies on stabilizing and then merging two aqueous streams of NPs in a nonpolar liquid containing complementary functionalized ligands. The resulting liquid threads consist of two distinct sections, each composed of specific groups of NPs. This unique duality inherent in Janus liquids affords an opportunity to assign distinct functionalities to opposing sides of the structures and fine-tune them independently. As a proof of concept and to highlight the future potential of these unique constructs, magnetic, conductive, or non-responsive NPs were sequestered in opposite sides of these constructs using magnetic graphene oxide (mGO)/GO, $\text{Ti}_3\text{C}_2\text{T}_x/\text{GO}$, or GO suspensions, allowing the formation of multi-responsive soft materials that can be heterogeneously tailored from the micro- to macro-scale.

Such precise control over the functionality of Janus constructs becomes even more pronounced when these structured liquids are used as templates, enabling the development of customized compositions and arrangements for task-oriented aerogels. The potential applications of Janus

aerogels are vast, especially when a deliberate connection is established between the functionality of the Janus building blocks and the ultimate application. For instance, the development of Janus aerogels with non-interfering magnetic/conductive opposing sections represents a breakthrough in electromagnetic interference (EMI) shielding. By integrating alternating magnetic/conductive domains within the aerogel structure, Janus constructs surpass existing shielding materials, offering superior wave absorption properties. This addresses a long-standing challenge in EMI shielding and positions Janus aerogels as highly promising candidates for next-generation shielding materials. This integration platform is also used to make functional aerogels flexible, where functionality, e.g., electrical conductivity, is customized through one side of the structures, and mechanical flexibility is mainly derived from the opposite side. This adjustable strategy opens numerous opportunities in pressure sensing and human motion monitoring.

We have revised the supporting information by adding the following sub-section:

Rationalization of Janus design: Controllable functionality and liquid templating concept

Janus liquids can be visualized as liquid threads consisting of two distinct sections, i.e., faces, each composed of a specific group of NPs. This duality embodies the essence of Janus, representing a remarkable opportunity to assign distinct functionalities to opposing sides of these structures and independently fine-tune them. This landmark nature of the Janus liquids becomes even more pronounced when they are used as a liquid template, enabling the development of customized compositions and arrangements for task-oriented aerogels.

The possible uses of Janus aerogels are limitless, especially when a deliberate connection is established between the functionality of the Janus building blocks and the ultimate application. For instance, developing Janus aerogels with non-interfering magnetic/conductive opposing sections is valuable for EMI shielding applications where alternating magnetic/conductive

domains of the aerogels create numerous interfaces within a small volume to absorb the electromagnetic waves (see the subsequent discussion in the following sections and Supplementary Information). One other application involves the development of Janus aerogels, where one section of the structure exhibits specific functionality, while the other imparts desired mechanical characteristics, as described in this paper. Overall, this anisotropic integration platform with adjustable functionality has many potential applications where the simultaneous encoding of multiple functionalities in a single construct is highly beneficial, e.g., Janus gas/liquid absorbers, where each section is fine-tuned to deal with a specific type of material, or multi-modal sensors that can accurately and independently sense two different classes of analytes.

3 -Improvement in the system (e.g., 3D printing, coaxial needles): As far as the suggestion of using co-axial needles: It was challenging to utilize coaxial needles to produce Janus systems. Specifically, we faced difficulties in characterizing the resulting structures due to the mixing of the aqueous suspension before and after exiting the needles. The primary reason behind this issue is the lack of viscoelastic properties in the suspensions used. As a result, upon contact, the suspensions readily mixed. Furthermore, we observed that the coaxial needle system was highly susceptible to gas bubbles, which occasionally appeared in the initial suspension. Besides, the utilization of Janus structured liquids in achieving predefined 3D geometries through 3D printing necessitates further advancements. It is imperative to acknowledge that the primary challenges in this field are closely associated with the limitations of commercially available 3D printers.

To successfully fabricate 3D-printed Janus structures while ensuring controlled processing of bulk Janus threads, it is crucial to employ two injection nozzles that remain spatially fixed in specific locations relative to each other throughout the printing process (please find further details in our response to Item #3 raised by Reviewer 1). Considering the existing commercial 3D printers, we

have encountered challenges when connecting the syringes containing the suspension to the injection head within a hexane bath using polymer-based tubes. These tubes are prone to movement due to even the slightest vibrations. In the case of the worm-like structures reported in our study, vibration does not pose a problem since both nozzles remain stationary, allowing us to secure them with clamps. However, for the purpose of 3D printing and achieving controllable processing, the use of metal tubing is necessary for fabricating Janus liquids.

4 - The bigger picture related to other liquid-liquid systems: Our efforts focused on reviewing previous studies on liquid-liquid systems and structured liquid threads, while also identifying their limitations. Thus far, the inherent characteristics of NPs in liquid systems have not been fully leveraged to achieve predefined applications. Additionally, traditional manufacturing methods for soft materials, including these liquid systems, have faced challenges in simultaneously tuning multiple functionalities within a single structure. Taking into consideration the previous comment, we have incorporated an additional section, i.e., Rationalization of Janus design: Controllable functionality and liquid templating concept, to highlight the most important characteristics of Janus structures that set them aside from other fabrication systems. We have added the following section in the main manuscript to provide background information regarding other liquid-liquid system and their limitations:

Liquid streaming is a new process where a jetted stream of one liquid in an external immiscible liquid can be stabilized by interfacial jamming, forming tubular liquid threads^{1, 5, 13}. For the efficient generation of these tubular all-liquid structures, the rate of NPSs formation, assembly, and jamming must be faster than the relaxation of fluids, while their binding energy must be high enough to impart mechanical stability^{14, 15, 16}. This process generates a tubular skin around the jetted liquid threads that maintains the integrity of this class of structured liquids^{1, 5, 15}. Taking

advantage of liquid jetting, several new concepts of structured liquids, e.g., all-liquid 3D printing^{15, 16, 17} and 3D water in oil tubular emulsions with tunable morphology and domain size⁵, were put forward. But, so far, little advantage has been taken of the combination of functional NPs and liquid streaming to generate constructs with a specific function in soft robotics¹⁸, microfluidics^{15, 19, 20}, or sensing^{17, 21}.

2. Using Janus term in the title and the text for structure and fluids is misleading and not descriptive of the system; perhaps the term multi or combined liquid streams can better describe the new system studied in this work. Since Janus particles are a defined term in materials science, using the term Janus for liquid/structure can be confusing. The same issue to the term "liquid streaming"; it is uncommon. It is recommended to use simpler, more common terminologies that are more descriptive of the systems.

Response: As the reviewer knows, the term "Janus" refers to the Greek/Roman god who embodies the paradoxical nature of beginnings and endings. Please note that the term Janus has been used in different contexts, beyond that of particle systems (electrospun fibers and others). Given that our structured liquids represent the integration of two contrasting functionalities, within a single construct, we deliberately chose to refer to them as "Janus liquids." Additionally, the term "liquid streaming or jetting" reflects our focus on stabilizing a continuous flow of liquid. Having patented these terminologies and utilized them in national and international outlets, we would like to keep the terminology as presented in the paper.

3. The use of programming for this system is overstated. Programming has different meanings in the field, and quantitative programming is not indicated in the current study.

Response: We agree with this point and the revised text replaces the words "programming" or "programmed" with alternative options that more accurately reflect the essence of our concept. Examples of such terms include "adjustable," "customizable," "versatile," and "encoding."

4. Page 13, line 225, it is not clear how the functionality of the systems is impacted by multi-scale porosity.

Response: We agree that it is necessary to rephrase the text to eliminate any ambiguity and misrepresentation. We originally used the term "multi-scale porosity" to highlight the high porosity of the aerogel, which spans from micro- to macro-scales and serves as one of the contributing factors to its internal scattering and EMI shielding. Our focus should be on the high porosity aspect. We now emphasize "high porosity ranging from micro- to macro-scales." Additionally, we have made the necessary adjustments in the abstract and other sections to ensure consistency.

Reviewer #3 (Remarks to the Author):

The following manuscript entitled "Functional Janus liquids" presents a study of preparing conductive lightweight hybrid of magnetic graphene oxide (mGO) and Ti_3C_2Tx MXene with Janus structure. The present work is interesting with very low reflection of EM waves; however, the authors have already published similar works in different journals (1. Small 2022, 18, 2200220; and 2. Advanced Materials Interfaces 2022, 9, 2101659, etc.). The present work is a total reproduction (with the same presentation of figures and results) of the works published by the authors before. In previous publications, authors used graphene oxide in liquid streaming to get

interfacial skin of GO in POSS-incorporated nonpolar (e.g., Hexane) solvent, whereas the current study includes MXene with the GO to obtain the same structure. Authors didn't provide any comparison of their own studies, that what makes a difference between GO/POSS and GO-MXene/POSS systems. Furthermore, the current manuscript lacks a sufficient understanding and explanation of the mechanisms behind the absorption characteristics of EMI shielding. Therefore, due to lack of novelty of the idea, this paper can't be published in prestigious journal of Nature Communications, and hence rejected.

Response: We appreciate the input provided by the reviewer but, we respectfully disagree with the assertion that our work is a "total reproduction" of our previous studies. We appreciate the opportunity to clarify this issue and to explain the distinctive nature and novel contributions of our research on Janus liquids.

The crux of our study revolves around the introduction of "Janus liquids" - a significant departure from our previous work. In the present paper, we pioneer the exploration of a stream of liquid endowed with two distinct and entirely separate regions, akin to two faces, allowing the precise control over the compartmentalization of distinct functionalities. We use the following images as an illustration that captures the essence of our work, showcasing a single stream of liquid divided into two immiscible sections, each possessing distinct functionalities.

Digital and EDX images of Janus liquids and aerogels introduced here for the first time. The visual evidence provided in this paper unequivocally attests to the distinctiveness and originality of our findings.

While it is true that we previously published a study in *Small* involving the manipulation of GO-based streams, it is important to emphasize that the concept of Janus liquids is an entirely distinct and original development. Janus liquids can be envisioned as liquid threads composed of two discrete sections, or faces, each containing distinctly different NPs. This intrinsic duality epitomizes the essence of the Janus concept, offering an extraordinary opportunity to bestow distinct functionalities upon opposing sides of these structures and independently fine-tune them. The pioneering nature of Janus liquids becomes even more pronounced when they serve as liquid templates, enabling the creation of tailored compositions and arrangements for task-oriented aerogels. The versatility and potential applications of Janus aerogels appear boundless, especially when a deliberate connection is established between the functionalities of the Janus building blocks and the ultimate application at hand.

We note that the use of a similar material, in this case, GO, as one component of the Janus structure does not warrant the assertion that the Janus concept represents a “total reproduction” of our prior

work. In our earlier study, we demonstrated the capability of GO to stabilize a falling liquid stream, whereas in this current research, we stabilize falling streams of magnetic and conductive nanomaterials, subsequently forming Janus structures. These advances, none of which were described in our previous work, underscore the distinctiveness and forward progress of our research. Furthermore, it is important to recognize that the adoption of similar processing techniques of GO does not compromise the novelty of our present paper. By that logic, papers focusing on 3D printing or other widely employed processing techniques would be deemed ineligible for publication, which is clearly not the case.

We note that there is no overlap when comparing the present work and our previous study in *Advanced Materials Interfaces*, since the latter addresses different aspects and introduce fundamentally different concepts.

Therefore, we appreciate the reviewer's feedback and have taken the comments into careful consideration. Our work is certainly novel and has significant interdisciplinary potential and academic value. A library of novel aspects of this research can be highlighted in the following table:

Innovation	Explanation	Future potentials
Development of Janus liquids	A stream of liquid endowed with two distinct and entirely separate parts, akin to two faces, allows for precise control over the assignment of distinct functionalities within a single liquid stream	Dual-encapsulation of cells and active matter, vessels for biphasic chemical synthesis and chemical separations, and all-liquid robots.
Janus liquids as templates for customizable aerogels	The landmark nature of Janus liquids becomes more pronounced when they are employed as a liquid template, enabling the development of customized compositions and arrangements for task-oriented aerogels.	Simultaneous encoding of multiple functionalities in a single construct. Multi-modal sensors and dual gas absorption.
Janus EMI shielding	Janus aerogels with non-interfering magnetic/conductive domains with groundbreaking EMI shielding absorption	
Enhancing mechanical characteristics of soft materials through the Janus concept. Application for human motion monitoring	Precise assignment of a functional material to one side of the Janus aerogels, while simultaneously stabilizing the entire structure with a specific group of nanomaterials on the complementary side.	Stabilizing aerogels and cryogels composed of nanomaterials that typically yield weak structures

Finally, in our effort to explore the EMI shielding characteristics of the Janus structures, we used a comprehensive approach. Our evaluation encompassed multiple aspects, including the consideration of high porosity spanning from micro-to-macro scales, the application of impedance matching/mismatching concepts, the interface between magnetic and conductive domains, and the inherent properties of nanomaterials. Furthermore, we have provided a comprehensive background on EMI shielding in the supplementary data, covering essential concepts within this field.

REVIEWER COMMENTS

Reviewer #2 (Remarks to the Author):

The comments are addressed.

Reviewer #3 (Remarks to the Author):

Manuscript #: NCOMMS-23-14122A

Recommendation: Major revision

In revision, the authors have commendably addressed a substantial portion of the raised comments and concerns. Nevertheless, there remain several inquiries to improve the manuscript's clarity and comprehensiveness.

1. What is the main advantage of Janus aerogel compared to the mixture of the magnetic strands and conductive strands? The reviewer suggests that the authors undertake a comprehensive comparison between the EMI shielding and sensing properties of Janus aerogel and a simple mixture comprising separate magnetic and conductive strands. This elucidation would facilitate an enhanced understanding of the advantages of Janus aerogel.
2. The absorption-dominating EMI shielding properties of Janus aerogels (with high total shielding effectiveness and low reflection coefficient) are very interesting. However, the authors have just intriguingly discussed the absorption-dominating EMI shielding attributes characterized by heightened total shielding effectiveness and a diminished reflection coefficient. Although the authors qualitatively mentioned about the impedance matching and porosity effects, a more profound comprehension of the shielding mechanism is hindered by the absence of raw data. For a comprehensive elucidation of the mechanism, it is advisable to conduct and present measurements encompassing porosity, electrical conductivity, dielectric permittivity, and magnetic permeability of Janus aerogels.
3. Additionally, for better understanding EMI shielding properties of Janus aerogel coated Cu film, EMI shielding properties of the pristine Cu film (0.1 mm) should be provided.

Reviewer's comments:

1. What is the main advantage of Janus aerogel compared to the mixture of the magnetic strands and conductive strands? The reviewer suggests that the authors undertake a comprehensive comparison between the EMI shielding and sensing properties of Janus aerogel and a simple mixture comprising separate magnetic and conductive strands. This elucidation would facilitate an enhanced understanding of the advantages of Janus aerogel.

Response: We thank the reviewer for the insightful comment. As a direct response to these suggestions, we have introduced an important addition to the article: an exploration of the EMI shielding characteristics of the samples comprising $\text{Ti}_3\text{C}_2\text{T}_x/\text{GO}$ (20 wt% GO), mGO/GO (50 wt% GO), and pure GO. Our objective here is to illustrate that, in the cases of mGO/GO (50 wt% GO) and pure GO, we encountered challenges in attaining effective EMI shielding characteristics due to their limited electrical conductivity and the inability of their structures to generate impedance mismatching characteristics. In contrast, with $\text{Ti}_3\text{C}_2\text{T}_x/\text{GO}$ (20 wt% GO) worm-like aerogels, distinguished by their high electrical conductivity, we observed reflection-dominated EMI shielding characteristics. This observation is further supported by the development of Janus EMI shields. The reviewer's comments, in conjunction with the newly added characterizations in the article, underscore the significance of Janus EMI shields in the context of our research.

Furthermore, we acknowledge the reviewer's points regarding the development of pressure sensors based on $\text{Ti}_3\text{C}_2\text{T}_x/\text{GO}$ (20 wt% GO), mGO/GO (50 wt% GO), and pure GO worm-like aerogels. However, in the case of mGO/GO (50 wt% GO) and pure GO samples, the absence of electrical conductivity precludes their suitability for sensing applications. Additionally, $\text{Ti}_3\text{C}_2\text{T}_x/\text{GO}$ aerogels, while exhibiting excellent electrical conductivity, lack compressibility and, therefore, are

unsuitable for pressure sensing due to their mechanical characteristics (**please see Figure S22 and Videos S9-12**). Indeed, one of the key advantages of Janus aerogels lies in their ability to enhance the mechanical properties of soft materials. Consequently, we can only develop effective pressure sensors based on Janus aerogels. In response to the comment, we have included the following section:

On page 14 of the main manuscript, we have:

Before discussing the absorption-dominant behavior of Janus aerogels, an examination of the EMI shielding characteristics of the worm-like structures composed of $\text{Ti}_3\text{C}_2\text{T}_x/\text{GO}$ (20 wt% GO), mGO/GO (50 wt% GO), and pure GO is conducted. It is noteworthy that both mGO/GO (50 wt% GO) and pure GO fail to exhibit the impedance mismatch characteristics crucial for effective EMI shielding, primarily due to their lack of electrical conductivity. Consequently, these structures are considered unsuitable for mitigating unwanted electromagnetic interference, as exemplified in Figures S23-25.

In contrast, the conductive worm-like structures of $\text{Ti}_3\text{C}_2\text{T}_x/\text{GO}$ (20 wt% GO) demonstrate an exceptional shielding effectiveness of 69.2 dB, a noteworthy achievement for lightweight electromagnetic shields (Figure S23). Importantly, this EMI shielding behavior is achieved without the need for thermal or chemical reduction processes. However, it is crucial to emphasize that the conductive properties of these structures, characterized by an electrical conductivity of approximately 120 ± 10 S/m, result in a high reflectance index of approximately 70%, corresponding to an absorptance of around 30%. This high reflectance introduces secondary reflection issues, a challenge that we endeavor to address through the development of Janus aerogels.

On pages 47-49 of the supporting information, we have:

Figure S23: (a, b) EMI shielding characteristics of the worm-like structures composed of Ti₃C₂T_x/GO (20 wt% GO), mGO/GO (50 wt% GO), and pure GO. (c) Electrical conductivity of Ti₃C₂T_x/GO (20 wt% GO) worm-like aerogels. The thickness of these aerogels is 3mm.

Figure S24: EMI shielding characteristics of pure GO aerogels, printed with a 200 μm nozzles. I and II refer to the data obtained from two different sides of the samples. The thickness of these samples was 3 mm. Three different samples were tested: (a, b) sample one, (c, d) sample two, and (e, f) sample three.

Figure S25: EMI shielding characteristics of mGO/GO (50 wt% GO) aerogels, printed with a 200 μm nozzles. I and II refer to the data obtained from two different sides of the samples. The thickness of these samples was 3 mm. Three different samples were tested: (a, b) sample one, (c, d) sample two, and (e, f) sample three.

On pages 17-18 of the main manuscript, we have:

Additionally, it is important to mention that the inclusion of GO on the opposite side of the Janus aerogels was aimed at improving the overall mechanical properties and compressibility, which cannot be achieved solely with the worm-like $\text{Ti}_3\text{C}_2\text{T}_x/\text{GO}$ (20 wt% GO), as shown in Figure S22 and Videos S9-12, along with their corresponding discussion.

2. The absorption-dominating EMI shielding properties of Janus aerogels (with high total shielding effectiveness and low reflection coefficient) are very interesting. However, the authors have just intriguingly discussed the absorption-dominating EMI shielding attributes characterized by heightened total shielding effectiveness and a diminished reflection coefficient. Although the authors qualitatively mentioned about the impedance matching and porosity effects, a more profound comprehension of the shielding mechanism is hindered by the absence of raw data. For a comprehensive elucidation of the mechanism, it is advisable to conduct and present measurements encompassing porosity, electrical conductivity, dielectric permittivity, and magnetic permeability of Janus aerogels.

Response: We thank the reviewer for the valuable insights. We have incorporated a comprehensive assessment of porosity spanning from macro- to micro-scales. To seamlessly integrate these evaluations into the flow of our results and discussion, we have woven them into sections dedicated to the fabrication of $\text{Ti}_3\text{C}_2\text{T}_x/\text{GO}$ worm-like aerogels and Janus structures.

Our thorough investigation delves into multiple factors, including the impact of needle size on macro-scale porosities, a detailed distinction between macro- and micro-scale porosities, and the influence of various types of nanomaterials on micro-scale porosities. These assessments have

been meticulously conducted using advanced techniques such as in-depth FESEM analysis, coupled with image processing and micro-CT tests.

The primary objective of these newly incorporated data is to shed light on the significance of and disparities between macro- and micro-scale porosities. This distinction sets our aerogels apart from previously reported porous structures. Consequently, these additional findings substantially enhance the coherence and relevance of our results and discussion, particularly in the context of the EMI shielding section.

Based on your comment, we also included data regarding the electrical conductivity of $\text{Ti}_3\text{C}_2\text{T}_x/\text{GO}$ worm-like aerogels, which were obtained using a four-point probe setup. We have combined this data with the EMI shielding characteristics of these structures, as discussed in previous comments from the reviewer. However, when it comes to evaluating the electrical conductivity of Janus aerogels, a different approach is needed, rather than using direct methods, like the four-point probe. Considering that Janus structures exhibit macro-scale heterogeneities, direct conductivity measurements with fixed pin positions result in significant variations, often represented by large-scale bars. Therefore, we propose that the conductance of these Janus structures should be indirectly assessed based on S11 and S22 factors obtained through Vector Network Analyzer (VNA) measurements, and subsequently evaluated in terms of reflectance and absorbance coefficients. We have extensively elaborated on this approach in the paper, and comprehensive datasets supporting this method are provided in the supplementary information. In summary, we conducted EMI shielding measurements for three different samples of each class of developed aerogels and assessed reflectance and absorbance across the X-band frequency range, ultimately calculating average values. These results, in our opinion, offer a more suitable means of presenting the electrical conductivity and elucidating the concept of impedance mismatch.

Regarding the dielectric permittivity and magnetic permeability of Janus aerogels, it's important to note that our current laboratory setup does not allow us to directly evaluate these specific parameters. Furthermore, we would like to emphasize that discussing these aspects falls slightly beyond the primary scope of our paper. The primary focus of our work is centered around the development of Janus structure liquids, and we have briefly touched upon potential future applications to inspire further research endeavors. We note that some studies have used software tools integrated with VNA setups to estimate dielectric permittivity and magnetic permeability parameters. However, we believe that for porous materials, especially those with significant thickness, the calculated parameters may not be entirely accurate. These software tools, including the version we have access to, were primarily designed for highly conductive and laminated structures, which are not in line with the unique characteristics of our constructs.

In response to the comment, we have included the following section:

On page 9 of the main manuscript, we have:

Figures S11-14 also demonstrate the potential for precise adjustment of the dimensions of aqueous tubes and the macro-scale voids between $\text{Ti}_3\text{C}_2\text{T}_x/\text{GO}$ tubules through a liquid streaming approach. As illustrated in Figures S11-12, the size of the extruded tubules can be varied from 450 to 850 μm by only using needles with different gauge numbers ranging from 21 to 15. Additionally, the presence of macro-scale voids among the $\text{Ti}_3\text{C}_2\text{T}_x/\text{GO}$ tubules is clearly evident in the cross-sectional X-Ray micro-computed tomography (micro-CT) images of these structures (Figures S13-14). The size and distribution of these macro-scale pores can be regulated by altering the dimensions of the extruded filaments. This remarkable flexibility in controlling the macro-scale porosity of $\text{Ti}_3\text{C}_2\text{T}_x/\text{GO}$ worm-like structures opens up exciting possibilities for diverse

applications, including electromagnetic interference (EMI) shielding, oil absorption, and gas absorption/detection, where precise control over macro-to-micro scale porosities is of paramount importance.

On page 11 of the main manuscript, we have:

The morphology of the Janus structures was investigated by FESEM and energy-dispersive X-ray spectroscopy (EDX) analysis and is shown in Figures 3e-n, S16 e-1, and S17-21. The Janus aerogels show dual porosities, which is the characteristic of worm-like aerogels, i.e., macro-scale porosities between the Janus tubes and micro-scale porosities within the tubules. Figures S17-18 illustrate the controllable manipulation of tube size and the subsequent connections and entanglements between tubules in Janus structures achieved by employing needles of different sizes during the streaming process. Additionally, to emphasize the micro-scale porosities within these aerogels, Figures S19-20 display FESEM images of hierarchical aerogels fabricated from pure GO ink, mGO/GO (20 wt% GO), and $\text{Ti}_3\text{C}_2\text{T}_x/\text{GO}$ (20 wt% GO) suspensions. It's noteworthy that the formation of these micro-scale pores results from the crystallization of ice during the freezing process, which is subsequently preserved upon freeze-drying. Furthermore, the heterogeneous distribution of NPs throughout the $\text{Ti}_3\text{C}_2\text{T}_x/\text{GO}:\text{mGO}/\text{GO}$ and $\text{Ti}_3\text{C}_2\text{T}_x/\text{GO}:\text{GO}$ tubules is evident in EDX analysis, where a distinct boundary on the surface of the Janus aerogels filament is observed (Figures 3e-n and S21).

On pages 37-39 of the supporting information, we have:

Figure S11: FESEM images of bulk and filaments of $\text{Ti}_3\text{C}_2\text{T}_x/\text{GO}$ (20 wt% GO) aerogels generated via liquid streaming approach using needles with different gauge numbers of (a) 21, (b) 18, and (c) 15. In the case of these samples, the extrusion pressure was fixed at 30 psi. Scale bars equal (I) 2 mm and (II-III) 1 mm.

Figure S12: Filament diameter of $\text{Ti}_3\text{C}_2\text{T}_x/\text{GO}$ (20 wt% GO) aerogels generated via liquid streaming approach using needles with different gauge numbers of (a) 21, (b) 18, and (c) 15. Average filament diameter of $\text{Ti}_3\text{C}_2\text{T}_x/\text{GO}$ (20 wt% GO) aerogels prepared at different needle gauges. In the case of these samples, the extrusion pressure was fixed at 30 psi.

Figure S13: Cross-sectional micro-CT images of $\text{Ti}_3\text{C}_2\text{T}_x/\text{GO}$ (20 wt% GO) aerogels prepared via gauge 15 needles. In the case of these samples, the extrusion pressure was fixed at 20 psi.

Figure S14: Cross-sectional micro-CT images of $\text{Ti}_3\text{C}_2\text{T}_x/\text{GO}$ (20 wt% GO) aerogels prepared via gauge 21 needles. In the case of these samples, the extrusion pressure was fixed at 20 psi. It's worth emphasizing that, when employing smaller needles for extruding filaments, the section of the sample situated nearer to the bottom of the holding vials demonstrates a denser composition with diminished macro-scale voids, as evident in images I to VI.

On pages 42-44 of the supporting information, we have:

Figure S17: FESEM images of $\text{Ti}_3\text{C}_2\text{T}_x/\text{GO}$ (20 wt% GO):GO Janus aerogels generated using needles with different gauge numbers of (a) 21 and (b) 18. Scale bars correspond to (I) 2 mm and (II-III) 1 mm.

Figure S18: Filament diameter distribution of $\text{Ti}_3\text{C}_2\text{T}_x/\text{GO}$ (20 wt% GO):GO Janus aerogels generated using needles with different gauge numbers of (a) 21 and (b) 18. (c) Average filament diameter of $\text{Ti}_3\text{C}_2\text{T}_x/\text{GO}$ (20 wt% GO):GO Janus aerogels prepared by needles with different gauge numbers.

Figure S19: Micro-scale porosity of the hierarchical aerogels made of (a) 10 mg/ml GO, (b) 10 mg/ml mGO/GO (50 wt% GO), and (c) 10 mg/ml $\text{Ti}_3\text{C}_2\text{T}_x/\text{GO}$ (20 wt% GO) suspensions at (I-II) 500 μm , and (III) 200 μm scale bars.

Figure S20: Micro-scale pore size distribution of aerogels made of (a) 10 mg/ml GO, (b) 10 mg/ml mGO/GO (50 wt% GO), and (c) 10 mg/ml Ti₃C₂T_x/GO (20 wt% GO) suspensions. Average micro-scale porosities of generated aerogels out of 10 mg/ml GO, mGO/GO, and Ti₃C₂T_x/GO suspensions.

3. Additionally, for better understanding EMI shielding properties of Janus aerogel coated Cu film, EMI shielding properties of the pristine Cu film (0.1 mm) should be provided.

Response: We thank the reviewer for the valuable insight. In response to this suggestion, we have included the EMI shielding characteristics of Cu tape in the supplementary information, specifically on pages 52-53.

Figure S28: Average EMI shielding characteristics of 0.05mm Cu tape.

Figure S29: EMI shielding characteristics of 0.05 mm Cu tape. Three different sections of the tape were tested: (a, b) sample one, (c, d) sample two, and (e, f) sample three.

REVIEWERS' COMMENTS

Reviewer #3 (Remarks to the Author):

NCOMMS-23-14122B

Recommendation: Accepted

In revision, authors have addressed all the raised comments and made substantial changes to improve the quality of the revised manuscript. The reviewer recommends this work for publication in Nature Communications.